# *Camellia japonica* Flowers as a Source of Nutritional and Bioactive Compounds

**DOI:** 10.3390/foods12152825

**Published:** 2023-07-25

**Authors:** Antia G. Pereira, Lucia Cassani, Chao Liu, Ningyang Li, Franklin Chamorro, João C. M. Barreira, Jesus Simal-Gandara, Miguel A. Prieto

**Affiliations:** 1Nutrition and Bromatology Group, Department of Analytical and Food Chemistry, Faculty of Food Science and Technology, Universidade de Vigo, Ourense Campus, 32004 Ourense, Spain; antia.gonzalez.pereira@uvigo.es (A.G.P.); luciavictoria.cassani@uvigo.es (L.C.); franklin.noel.chamorro@uvigo.gal (F.C.); mprieto@uvigo.es (M.A.P.); 2Key Laboratory of Novel Food Resources Processing, Ministry of Agriculture and Rural Affairs/Key Laboratory of Agro-Products Processing Technology of Shandong Province/Institute of Agro-Food Science and Technology, Shandong Academy of Agricultural Sciences, Jinan 250000, China; liuchao555@126.com; 3College of Food Science and Engineering, Ocean University of China, Qingdao 266005, China; ningyangli@126.com; 4Centro de Investigação de Montanha (CIMO), Instituto Politécnico de Bragança, Campus de Santa Apolonia, 5300-253 Bragança, Portugal

**Keywords:** camellia, petals, nutritional characterization, lipids, thermal analysis, nutraceuticals

## Abstract

In recent decades, plants have strengthened their relevance as sources of molecules potentially beneficial for health. This underpinning effect also arises from the extensive research that has been conducted on plants that are typically undervalued, besides being scarcely used. This is the case with Camellia japonica in Galicia (NW Spain), where, despite its abundance, it is exclusively used for ornamental purposes and has been studied only for its proximate composition. Thus, the present study was conducted on several additional parameters in the flowers of eight *C. japonica* varieties. Our results show that camellia has a high nutritional value, with carbohydrates as the most abundant macronutrients followed by a moderate protein content (4.4–6.3 g/100 g dry weight) and high levels of polyunsaturated fatty acids (especially ω-3 fatty acids, which represent 12.9–22.7% of the total fatty acids), raising its potential for use for nutritional purposes. According to the thermochemical characterization and elemental composition of camellia, the raw material has poor mineralization and low nitrogen content, but high percentages of volatile matter and high carbon-fixation rates, making it a promising alternative for biofuel production. Furthermore, preliminary analysis reveals a high concentration of different bioactive compounds. As a result of these findings, camellias can be used as food or functional ingredients to improve the nutritional quality of food formulations.

## 1. Introduction

Famine and/or poor nutrition, combined with health problems, has emerged as one of the world’s most serious issues [1]. Malnutrition or unbalanced diet can cause anemia (iron deficiency), mental impairments (iodine deficiency), blindness (vitamin A deficiency) [2], or even chronic diseases (e.g., tumor, cardiovascular diseases, or diabetes) [3,4]. 

Plants have been integrated into human diets since time immemorial, mostly due to their nutritional value and wide acceptability. In recent years, plants have shown, as well, beneficial health properties, promoting alternative applications (e.g., gastronomy and cosmetics) [5]. However, there is a considerable number of underappreciated and underutilized plant species, despite their potential to diversify human diets and increase food production levels. Because many essential food products are inaccessible to the entire population (primarily due to economic constraints) [6], using underexplored plants promotes more eco-friendly and resilient agri-food models [7,8], improving the economic development of flower- and plant-producing regions [9]. Despite recent trends, such as using flowers in modern cuisine or as novel ingredients, few studies on their nutritional composition have been conducted to date, particularly regarding their micronutrient composition. *Camellia* sp. (family Theaceae), a subtropical evergreen tree with high nutritional and medicinal value, is such an example. Despite more than 120 camellia species existing [10], the most common are *C. sinensis* (tea production), *C. oleifera* (oil production) and *C. japonica* (ornamental flowers) [11]. *Camellia* species are mainly found in Asia (specifically, Japan, China, Vietnam, and India) [12], although they are also found in Western countries [13]. *Camellia japonica* L., in particular, is a prevalent species established in Galicia (northwest Spain), one of the global reference markets for its cultivation and production [14]. In fact, Spain grows approximately 2.5 million camellia shrubs per year (mainly *C. japonica*), which are exported throughout Europe for ornamental purposes [15]. Despite its unquestionable use as an ornamental bush, little effort has been made to characterize the health-promoting properties of *Camellia japonica* L., as well as its efficacy as a source of nutritional and bioactive compounds [11]. 

Considering its high water content (>87%), the drying process of *C. japonica* flowers is a critical step for the quality and value of camellia flower products [16,17,18]. On a dry weight basis, *C. japonica* contains significant amounts of protein, total dietary fiber, and minerals [18], together with considerable percentages of triterpenes, glycosylated flavonoids, saponins, and tannins, which are widely known for their biological properties such as antioxidant, anti-viral, antifungal, and/or cytotoxic activity [19]. Therefore, *C. japonica* must be accounted for as a potential source of nutrients and high-value molecules.

In this work, the nutritional and bioactive compound composition of eight varieties of *C. japonica* L. flowers (Figure 1) was evaluated. The chemical characterization of these underexplored flowers highlights their potential applications as functional ingredients in food, nutraceutical, and cosmetic industries. 

## 2. Materials and Methods

### 2.1. Sample Collection

The flowers used in this study were obtained from eight different cultivars of *Camellia japonica* L. These cultivars were botanically identified by official germplasm banks and identification guide resources. The specific cultivars include in this study were ‘Conde de la Torre’ (CT), ‘Elegans variegated’ (EV), ‘Donation dentada’ (*Camellia japonica* × *Camellia saluenensis,* DD), ‘Dr. Tinsley’ (DT), ‘Eugenia de Montijo’ (EM), ‘Grandiflora superba’ (GS), ‘Hagoromo’ (HA), and ‘Carolyn Tuttle’ (CT). These cultivars were selected for their distinct characteristics and were used in subsequent analyses and experiments. The collection took place in NW Spain (42.431° N, 8.6444° W) in January 2020, and the collection was carried out by Viveiros Moreira. After collection, flowers were frozen at −80 °C and lyophilized (LyoAlfa10/15, Telstar, Thermo Fisher Scientific) under specific conditions: the freezing time in the equipment was set to 1 h, followed by a warming process for 20 min using a back-vacuum (BV) method. The main drying phase lasted for 2 h and 30 min at a pressure of 0.01000 mbar, while the final drying stage extended for 72 h at a pressure of 0.0010 mbar. Once lyophilized, the flowers were crushed into a fine powder (1.8 mm mesh) and stored at −20 °C until analysis.

### 2.2. Evaluation of Camellia as a Promising Source of Nutrients

The proximate analysis (moisture, ash, fat, and dietary fiber) of each flower variety was performed according to the AOAC protocols [20] and expressed as g/100 g dry weight (DW). All determinations were analyzed in triplicate. 

#### 2.2.1. Lipid Content

Samples (2 g) were placed in a flask containing 100 mL of n-hexane in an automatic Soxhlet apparatus (Büchi Labortechnik AG, Flawil, Switzerland). The extractions were performed at 68 °C during 3 h. The extract was then concentrated (complete solvent removal) using a rotary evaporator. The lipid content was determined gravimetrically, and the results were expressed as g/100 g DW.

##### Fatty Acid Characterization

To obtain volatile fatty acid methyl esters (FAMEs), the oil sample obtained with Soxhlet extraction (Section 2.2.1) was submitted to an esterification/transesterification reaction [21]. In brief, the lipid phases were resuspended in 1 mL of toluene and 2 mL of sulfuric acid (1% in methanol), and the mixture was incubated (50 °C) overnight. Afterwards, samples were allowed to cool protected from light. Then, 5 mL of a 5% sodium chloride solution were added. The resulting mixture was stirred (30 min), further added to with 5 mL of hexane, and stored in the dark until two layers were formed. The supernatant was recovered and added to with hexane (5 mL). Once the FAME-containing supernatants were obtained, they were pooled, and 4 mL of water (2% sodium bicarbonate) was added before being frozen overnight. The liquid phase (hexane) was recovered and frozen until analysis.

A gas chromatograph (Chromatographic System Agilent 7820A) equipped with an Agilent HP-88 (60 m × 250 µm × 0.25 µm) column and a flame ionization detector was used to analyze the FAME-containing extracts. The chromatographic method employs two temperature programs: the first consists of a ramp of 5 °C/min until reaching 220 °C, holding for 15 min. The second uses a temperature ramp of 40 °C/min until reaching 250 °C, holding this temperature for 2 min. Helium was used as the carrier gas (1 mL/min). Finally, FA identification was performed using external standards and FA composition was determined using corresponding calibration curves. The results were expressed as % of total fatty acid content.

#### 2.2.2. Protein Content

Total protein content was determined using the Dumas technique [22], as described in AOAC protocols [23]. Briefly, samples (about 5 mg) were combusted at high temperatures (flash combustion), resulting in total and immediate oxidation of organic compounds. A series of combustion gases were produced during the process (O_2_, CO_2_, H_2_O, N_2_, and nitrogen oxides). These gases were collected and filtered to remove O_2_, CO_2_, and H_2_O. Then, the nitrogen oxides were reduced to nitrogen at high temperatures in a copper reduction column. Pure helium was used as the mobile phase. Total nitrogen (including both inorganic and organic fractions) was determined using gas chromatography (Elemental Analysis Unit FISONS Carlo Erba EA1108, with a thermal conductivity detector). The detection limit was set to 10 ppm. Plant proteins have an average nitrogen (N) content of about 16%, so nitrogen content is multiplied by 6.25 (1/0.16) to obtain protein content. The results were reported as g/100 g DW.

#### 2.2.3. Carbohydrate Content (Simple Sugars by Difference)

Carbohydrate content was estimated assuming a 100 g dried sample reference, calculating the difference among this value and the sum of all the other components (lipid, protein, and ash) [24].

#### 2.2.4. Dietary Fiber Content

Insoluble and soluble dietary fiber contents were determined using the AOAC 991.43 enzymatic–gravimetric procedure [23]. Briefly, 1 g of sample was thermo-digested sequentially with α-amylase, protease, and amyloglucosidase to digest starch and protein.

Then, the insoluble dietary fiber was strained, and the remaining residue was washed with distilled water and further precipitated with ethanol (95%). The precipitate was recovered, filtered, and dried before being analyzed for soluble and insoluble dietary fiber values. These values were corrected with the protein, ash, and blank data [25,26]. Total dietary fiber was calculated as the sum of soluble and insoluble fiber [27].

#### 2.2.5. Macro- and Micromineral Content

The content of microelements (iron (Fe), manganese (Mn), copper (Cu), zinc (Zn), and molybdenum (Mo)) and macroelements (calcium (Ca), potassium (K), magnesium (Mg), sodium (Na), and phosphorus (P)) was determined by inductively coupled plasma optical emission spectrometry (ICP-OES). A Perkin–Elmer Optima 4300 DV spectrometer (Shelton, CT, USA), equipped with an AS-90 autosampler axial system, a high dynamic range detector, and a cross-flow-type nebulizer for pneumatic nebulization was used [28]. All results were expressed as mg/kg.

The ICP-OES analysis followed the method described in previous articles [29]. In short, 0.25 g of each sample was digested with nitric acid (HNO_3_) and hydrogen peroxide in a Multiwave 3000 oven (Anton Paar, Graz, Austria), which was equipped with eight digestion vessels. Following that, elements were examined by ICP-OES (axial configuration). The experimental conditions of the equipment were those reported previously by Millos [29]. The calibration curve was created using a stock solution using ^115^In as internal standard.

The concentrations of the microelements selenium (Se) and iodine (I) were simultaneously analyzed by inductively coupled plasma mass spectrometry (ICP-MS) using a Thermo Elemental X7 Series ICP-MS equipped with an ASX-520 autosampler (Omaha, NE, USA) and PlasmaLab Software (version 1.5). Before ICP-MS analysis, 0.25 g of each sample was digested using tetramethylammonium hydroxide (TMAH). Both the ICP-MS measurement and the operating conditions were carried out in accordance with Millos et al. [30]. All results were expressed as mg/kg.

#### 2.2.6. Moisture and Ash Content

The moisture content of the flowers was determined thermogravimetrically in accordance with the standardized protocol UNE-EN 14774-1 [31]. Briefly, 250 mg of each sample was placed in an oven at 105 °C under an inert nitrogen flow (30 mL/min) until a constant weight was reached. The equipment used in the procedure was a SETSYS Evolution (Setaram Instrumentation, Caluire-et-Cuire, France).

The ash content was determined thermogravimetrically in accordance with the standardized procedure UNE-EN 14775 [32]. Briefly, 7–15 mg of water- and volatile-matter-free samples were ignited at 900 °C beneath a flow of an O_2_-rich gas (30 mL/min) until constant weight.

#### 2.2.7. Basic Elemental Composition (C, O, N, H, S)

The carbon (C), nitrogen (N), hydrogen (H), and sulfur (S) contents of the raw material were determined using the same equipment as in Section 2.2.2 [28,33]. The oxygen (O) concentration was set up by difference using the data obtained from the remaining element concentrations.

#### 2.2.8. Thermal Analysis

For thermochemical characterization of *C. japonica* flowers, non-isothermal analyses were conducted by heating samples (7–15 mg) in a temperature range of 20–900 °C at 10 °C/min. Nitrogen was selected as the mobile phase (30 mL/min). Isothermal experiments were conducted at 900 °C with an O_2_-rich gas flowing (30 mL/min). Throughout the thermal analysis, the sample weight, furnace temperature, mass loss (%), mass loss rate (%/min), and heat flow (mW) were recorded [34]. All assays were performed in a SETSYS Evolution thermal analyzer (Setaram Instrumentation, France).

### 2.3. Determination of Bioactive Compounds from C. japonica L. Flowers

Heat-assisted extraction was performed in accordance with previous studies [35] with some modifications. A 2 g amount of lyophilized flowers was placed in amber glass bottles with 50 mL of methanol (60% *v*/*v*; solid/liquid ratio 40 g/L). Once the bottles were filled, they were incubated at 50 °C for 1 h in a thermostatic water bath under continuous agitation. The obtained mixtures were then centrifuged in Falcon tubes (6037× *g*-force, 15 min) and supernatants were collected and further analyzed.

#### 2.3.1. Extraction Yield

Extraction yield was thermogravimetrically measured in accordance with previous research [36], calculated using Equation (1) and reported as g extract/100 g DW. Determinations were performed fivefold.
(1)$$\mathrm{Yield}\;\left(\%\right)=\frac{P_{t=24\;h}-P_{t=0}}{\left(\frac{m_{\mathrm{sw}}\times V_{a}}{V_{\mathrm{sv}}}\right)\times \left(\frac{100-{\mathrm{MC}}_{\mathrm{sw}}}{100}\right)}\times 100$$
where P_t=0_ is the mass of the crucible before adding the extracted solution; P_t=24h_, mass of the crucible after 24 h of drying; m_sw_, mass of the dry sample; V_a_, volume of extracted solution aliquot (5 mL); V_sv_, volume of solvents used for extraction (50 mL); and MC_sw_ the moisture content (%) of each sample.

#### 2.3.2. Total Phenolic Content (TPC)

TPC was determined spectrophotometrically as described by Cassani [37]. Briefly, several serial dilutions were prepared, beginning with a dilution of 1/32 and ending with a dilution 1/512. To each 25 µL of each of these dilutions, 125 µL of FCR (diluted in water, 1:10) was added. After 3 min of incubation at room temperature, 100 µL of sodium carbonate (Na_2_CO_3_) solution (7.5% *w*/*v*) was added and the reaction mixture was incubated for 2 h (room temperature). A microplate reader was used to measure absorbance at 765 nm. Gallic acid concentrations ranging from 5 to 100 µg/mL was used as standard. Results were expressed as mg gallic acid equivalents (GAE)/g DW. Determinations were conducted fivefold.

#### 2.3.3. Total Carotenoid Content (TCC)

TCC was calculated using spectrophotometry. The absorbances of the different extracts were measured at 450 nm using a microplate reader [38]. TCC was calculated using Equation (2) proposed by Scott [38]. Results were expressed as µg total carotenoids/g DW. Determinations were conducted in triplicate.
(2)$$\mathrm{TCC}=\frac{A_{450}}{2500}\times 10\frac{\mathrm{mg}}{\mathrm{mL}}\times \frac{V_{\mathrm{sv}}}{m_{\mathrm{sw}}\times \frac{100-{\mathrm{MC}}_{\mathrm{sw}}}{100}}$$
where A_450_ is the absorbance measured at 450 nm; m_sw_, mass of dry sample; V_sv_, volume of solvents used for extraction (50 mL); and MC_sw_ the moisture content (%) of each sample.

#### 2.3.4. Total Flavonoid Content (TFC)

TFC was spectrophotometrically determined using the methodology proposed by Zhishen et al. [39] with some modifications. Briefly, 1/8 dilutions of the extracts were prepared. A 100 µL volume of each dilution was transferred to a Khan tube, which was then filled with 400 µL of distilled water and 30 µL of sodium nitrite (5% *m*/*v*). After 5 min of room-temperature incubation, 30 µL of aluminum chloride (10%) was added, followed by 200 µL of sodium hydroxide (1M) and 240 µL of distilled water (after 1 min). The reaction mixture was immediately transferred to a microplate after being stirred in a vortex. Absorbance was measured at 510 nm using a microplate reader. The results were expressed as µg quercetin/g DW. Determinations were conducted in triplicate.

#### 2.3.5. Total Anthocyanin Content (TAC)

TAC was determined by the pH difference method [40]. Briefly, 50 µL of each sample was dissolved in 200 µL of two buffer systems: potassium chloride buffer (0.025 M; pH = 1) and sodium acetate buffer (0,4 M; pH = 4.5). Absorbance was measure at 510 nm and at 700 nm. TAC was calculated using Equation (3). Results were reported as µg cyanidin-3-glycoside/g sample.
(3)$$\mathrm{TAC}=\frac{{\left(A\times \mathrm{Mw}\times \mathrm{DF}\times 1000\right)}_{}}{\varepsilon \times 1}$$
where Mw is the molecular weight cyanidine-3-glucoside (449.2 g/mol); DF, dilution factor (5); and ε the molar absorptivity (26,900 L/mol).

A: Equation (4)
(4)$$A={\left(A_{510}\;–{\;A}_{700}\right)}_{\mathrm{pH}\;1.0}-{\left(A_{510}\;–{\;A}_{700}\right)}_{\mathrm{pH}\;4.5}$$
where A_510_ is the absorbance measured at 510 nm and A_700_ the absorbance measured at 700 nm.

### 2.4. Statistical Analysis

Results were expressed as mean values with standard deviation (SD). XLSTAT version 2021.4 (Addinsoft, Paris, France, 2021) and R software version 4.2.0 (R Development Core Team, 2011) were used to evaluate data. Analysis of variance (one-way ANOVA, *p* < 0.05) and Tukey–Kramer comparison test were performed to study the effect of different varieties of *C. japonica* flowers on extraction yield, TPC, TCC, TFC, and TAC.

## 3. Results and Discussion 

### 3.1. Nutritional Composition of C. japonica Flowers

#### 3.1.1. Lipid Content and Fatty Acid Profile

Table 1 shows the chemical composition of *C. japonica* L. flowers. The lipid content of the camellias studied ranged between 0.85 and 1.55 g/100 g DW, with Donation dentada (DD) having the highest content. These values are higher than those obtained by Fernandes [18], who reported a lipid concentration in *C. japonica* flowers of 0.31 g/100 g DW. Similar lipid contents were reported by Kim (0.51–0.82 g/100 g DW) [41]. On the other hand, the total lipid content of *C. japonica* L. flowers is lower in comparison to other parts of the plant, such as the seeds, where the lipid content ranged between 16.1% and 31.9% [42]. However, according to some studies, some varieties of *C. japonica* can be used as oilseed crops after selecting genotypes with high oil and high oleic acid content [43].

According to Table 2, the flowers of the studied Camellia varieties in this present study exhibit a content comparable to other plants and higher than that of mushrooms. However, their content is lower than that of well-known sources of healthy fats, such as nuts, with a content up to 58 times lower than almonds.

Table 1 shows the fatty acids (FAs) profile of the camellias under study. Polyunsaturated fatty acids (PUFAs) are the main contributors to the total FA content in all of the examined varieties with percentages ranging from 61 to 71% for Dr. Tinsley (DT) and Carolyn Tuttle (CT) varieties, respectively. This percentages of PUFAs are significantly higher than other crops (Table 2). Linoleic acid (C18:2 ω-2) was the most abundant compound in all varieties of *C. japonica* under study, with no significant difference (*p* > 0.05) among samples. This concentration is significantly higher than that of other reference oils such as olives (7.4%), peanut (21.6%), and soybean (50%) [15]. This is significant because all isomers of linoleic acid have bioactivity, including anticancer, antiobesity, antidiabetic, and antihypertensive properties [53]. In addition, it is an essential omega-6 FA that is used as a functional ingredient in many food industry products [54]. The camellias under study can be considered a source of balanced omega FAs based on the observed ω6/ω3 ratio values, because they are in the recommended range (ω6:ω3 2:1) [55].

*C. japonica* flowers studied herein had lower levels (3.79 to 4.72%) of monounsaturated FAs (MUFAs) than those reported in other studies (25%) [18]. Oleic acid (C18:1) was the most prevalent MUFA in all the varieties analyzed, accounting for 2–4% of total fatty acids. This FA is found in higher concentrations in the oil extracted from *C. japonica* seeds (86.3% of neutral lipids, with neutral lipids accounting for 88.2% of the sample lipid fraction) [56,57]. However, these *C. japonica* flowers varieties can be considered a relevant source of *cis*-11-eicosenoic acid (0.45%).

The total saturated fatty acid (SFA) concentration is consistent with previous studies that found 45% of SFAs in *C. japonica* flowers [18]. This concentration is greater than that found in other parts of the plant. For example, SFAs account for 12.39% of total lipids in cold-pressed oil from *C. japonica*. *C. japonica* varieties under study had higher SFA levels than common oils (e.g., 12% olive and sunflower oil, 15% soybean, 16.6% peanut) [15]. According to our findings, palmitic acid is the most abundant SFA in *C. japonica* flowers, with concentrations ranging from 15.97 to 25.52% and no significant differences between varieties. These levels are higher than those found in *C. japonica* seed oils (14.5%) [58]. Myristic acid concentration reached the highest value (1.73%) in Grandiflora Superba (GS), while Conde de la Torre (CT) and TU contained the lowest (0.79 and 0.72%, respectively), showing generally higher percentages than in previous studies. For example, Fernandes reported 0.43% myristic acid concentrations in *C. japonica* flowers [18]. Similar comparisons can be made with stearic acid (3.09–1.73%), which is found in the literature at concentrations up to three times lower (0.86%) [18]. In facst, the results obtained in this study (≈2%) are comparable to those obtained with other parts of *C. japonica*, such as seeds (1.4%) [59], which can produce 12.1% stearic acid [58]. On the other hand, *C. japonica* flowers had a higher concentration of arachidic acid (1.44–2.47%) than oil obtained from other camellias species (e.g., *C. oleifera* 0.22–0.70%) [60]. 

#### 3.1.2. Protein Content

The protein content of the camellias studied ranged from 4.43 to 6.34 g/100 g DW with no statistically differences (*p* > 0.05) for most cases (Table 1). Fernandes et al. reported a protein content of 0.76 g/100 g fw (fresh weight) in *C. japonica* flowers [18]. These levels are higher than those found in *C. japonica* leaves, which contain less than 0.01 g/100 g DW and 90 µg/g DW of soluble protein [61]. Protein concentration is even higher when a purified preparation of the callose plug from *C. japonica* pollen tubes is analyzed, with values reaching approximately 24% of DW [62]. However, the protein content of the flowers varies significantly, depending on their maturity stage. For example, *C. sinensis* flowers had higher levels of proteins in stages 1 (unopened bud) and 2, decreasing with flower development [63]. Because flowers used in this study were at the end of their maturation, the protein contents may be similar to the rest of the plant at earlier stages of maturation. In fact, in other species of camellia (*C. sinensis*), the concentration in flowers was found to be 2–3 times higher than in the leaves (15 and 30–50 g/100 g DW) [64]. Other camellia species such as *C. nitidissima* had a protein content (≈7.4%) comparable to our results, with an essential amino acids composition comparable to the FAO/WHO reference protein [65]. Tea plant flowers (*C. sinensis*) have a protein content of 4.05% [66]. Other studies have found that the main free amino acid compounds in *C. japonica* flowers are histidine, aspartic acid, and glutamic acid, whereas the major amino acids are histidine and alanine. The concentration of total free amino acids and total amino acids decreases as picking time is delayed, whereas the ratio of essential amino acids to total amino acids increases [41]. 

#### 3.1.3. Mineral Content

According to Table 3, results indicate that all studied camellias are a good source of nutritionally important minerals like Ca, K, Mg, P, Na, Fe, and Mn in comparation with different industrial crops (i.e., Table 2). DT had the highest mineral content, with 1156.0 mg/kg Mg, 419.5 mg/kg Na, and 9.580 mg/kg Zn. Hagoromo (HA) had the highest contents of Ca (2544.5 mg/kg) and P (1536.5 mg/kg), which was similar in Elegans variegated (EV: 1566.0 mg/kg). K content was higher in all samples than sodium, as observed by Ferrara in *C. sinensis* [67]. No significant difference were found between camellia varieties for Se content. The macromineral concentration was higher than that found in conventional vegetables (e.g., rye, barley, beans) [68]. Mo and I content were found to be below the quantification level which was consistent with Ferrara’s findings [67]. The mineral content of *C. japonica* flowers decreases in the following order: K > Ca > P > Mg > Na > Mn > Fe > Zn > Cu > Se. Other camellia species (*C. nitidissima*) have similar mineral content in the leaves, with high levels of N, K, Ca, and Mg, and lower contents of Zn (N > K > Ca > Mg > P > Fe > Zn > Mo) [65]. The mineral concentration may vary depending on sample collection time. Harvesting at a later stage may increase the minerals content of *C. japonica* (P, Ca, K, Na, and Fe) [41]. However, no significant differences in mineral contents (N, P, K, Ca, Mg, and Zn) were found between cultivated and wild camellia plants [65].

### 3.2. Thermochemical Characterization of Camellias

Table 3 shows thermal analysis and elemental composition of the camellias studied. Thermal analysis revealed that moisture contents ranged from 3.82% for EV to 9.91% for GS. These results are comparable to those reported for other lyophilized flowers (5.32–12.38%) [69,70]. The moisture content of fresh *C. japonica* flowers can reach 87.7% [18]. 

There were no significant differences in ash content between camellia varieties (1.75–3.80%) (Table 3). Previous studies reported an ash content of 0.37 g/100 g fw in *C. japonica* flowers [18]. These results are lower than those obtained from other biomasses used in combustion processes (e.g., 7% in grass biomass, 0.1% in wood chips, 17.2% in cotton gin trash, 6% in alfalfa seed straw, 16–23% in rice hulls, or 5–17% in coal) [71,72] and other edible flowers (e.g., 8.07% in *Aloe vera* flowers or 9.17% in sunflowers) [73,74]. Camellia flowers could be used as an alternative energy source in this context because their ash content was less than the maximum established in pyrolysis processes (20%) [75].

The volatile content of camellia flowers ranges from 65.03% (GS) to 73.53% (EV) (Table 3). These values are comparable to the majority of biomass (65–85%), but lower than woody biomass (76% to 86%) [75]. Thus, camellia flowers are susceptible to thermal degradation during pyrolysis due to their high volatile matter contents [75,76]. All camellias studied meet two important requirements for biofuel use: high volatile matter content and low ash content. As a result, camellias may be a viable biofuel production option. EV had the lowest ash content and the highest volatile matter content of all varieties studied, indicating a potential alternative use in biofuel production. 

The proportion of fixed carbon (Table 3) in the camellias studied ranged from 19.74–22.23%. These results are consistent with previous findings for other plant-based materials [75] and biomass wastes [77]. 

The high heating value (HHV) of camellia flowers under study ranged from 21.17 to 22.33 MJ/kg (Table 3), which were higher than those found in other floricultural wastes (e.g., 18.520 kJ/kg dry stems of roses; 18.030 kJ/kg sunflowers; 15.560 kJ/kg dried chrysanthemums; 15.210 kJ/kg tulips) [78]. Therefore, camellia flowers are a better substrate for combustion than other floricultural waste and comparable to other crop byproducts (20.80 kJ/kg) [79].

The elemental composition (Table 3) is similar to that reported by Lee for *C. japonica* oil cake (48.6% C; 7.5% H; 1.2% N), but with higher levels of O (≈48% vs. 39.80%) [80]. Therefore, camellia flowers had similar carbon and hydrogen proportions to those reported for other plants. In general, biomass has a carbon content of about 45% and a hydrogen content of 6.5% [81]. The nitrogen content (0.71–1.02%) was lower than in common agro-industrial wastes (e.g., bean husk: 2.64%, coffee pods: 2.53%, wheat bran: 2.34%), or sugar cane bagasse (0.89%). This proportion of nitrogen is desired if the goal is to produce biofuel because it allows for the reduction of nitrogen oxide (NOx) emissions [82]. Therefore, camellias could be used to produce biofuels. However, *C. japonica* L. flowers are a rich source of bioactive compounds [11]; thus, their extraction can be a good opportunity for adding value. Using camellia flower byproducts for fuel production is the most viable option for making camellia biofuel a profitable process. In this way, the flowers could be used for the extraction of bioactive compounds and nutritional purposes and the extraction process residue could be used to produce biofuel. This system could reduce costs, while also attenuating environmental impact [83] and having no competition for food production (e.g., corn or algae not used for biofuel production).

### 3.3. Evaluation of Camellia as a Promising Source of Bioactive Compounds

Table 4 shows the extraction yield, TPC, TCC, TFC, and TAC of the camellia extracts studied. Significant differences in the extraction yield (*p* < 0.05) were found among the different camellia varieties ranging from 55.14 to 62.68%, with GS, EV, Eugenia de Montijo (EM), HA, and TU exhibiting the highest extraction yields. 

This could be due to an increase in the release of polar-soluble molecules, such as polysaccharides, proteins, peptides, or organic acids. On the other hand, DD exhibited the lowest extraction yield (55.14%). These values correspond to the nutritional composition because DD was the camellia with the highest lipid content. All the varieties have consistent yields, which can be attributed to the use of intermediate-polarity solvents. Therefore, the use of heat-assisted extraction in conjunction with aqueous methanolic extracts results in an efficient method for extracting bioactive molecules from *C. japonica* flowers.

Regarding phenolic compounds, TPC ranged between 78.45 and 108.64 mg GAE/g DW (Table 4). No significant differences (*p* > 0.05) were detected between varieties. Nonetheless, the comparison to other results is an arduous task because different extraction methods are used (solvent, temperature, etc.). There are also differences in how results are expressed, with the most common being mg GAE per g extract rather than mg GAE per g DW. Other studies on *C. japonica* flowers found lower concentrations that depended on the color of the flower (4.8 mg·GAE/g DW for white, 6.2 mg·GAE/g DW for red, and 19.6 mg·GAE/g DW for pink) [84]. Similar results were obtained by Chen et al. (10.45 mg GAE/g DW) [85]. Trinh et al. found 56.7–107.6 mg GAE/g DW in camellia flower extracts after enzymatic treatments [86]. Li reported 5.14 mg GAE/g FW in *C. japonica* leaves [87]. Similar findings were obtained by Páscoa et al., who reported 5.7 mg GAE/g dry leaf [88]. Higher concentrations (9.06 mg GAE/g DW) were found in the calli of *C. japonica* [89]. These differences in TPC are due to the high dependence of these compounds on the maturity, seasonality, and geographic and climatic factors. As a result, *C. japonica* young leaves contain higher TPC (74.62 mg/100 g DW) than flower buds and flowers (65.02 mg/100 g DW and 62.42 mg/100 g DW, respectively) [90]. The main phenolic groups found in *C. japonica* are phenolic acids, flavonoids, and tannins, with contents varying depending on the evaluated plant part, with flowers and leaves being the most abundant [11]. According to the results of this study, the extraction yield may not be strongly related to TPC of camellia, as significant differences were found. Therefore, other chemical constituents (e.g., sugars) may be present in the obtained extracts [91]. 

Concerning total carotenoids, significant differences were found between camellia species, ranging from 26.03 to 73.21 µg carotenoids/g DW (Table 4). DT, HA, and GS had the lowest values (<30 µg/g DW), while EM showed the highest TCC (>70 µg/g DW). These values are in agreement with the color of the flowers, with pink flowers having the highest TCC and white ecotypes having the lowest TCC. Fernandes reported higher TCC (0.247 mg/g DW) in *C. japonica* flowers [18]. This higher value was ascribed to the optimal degree of maturation of flowers. TTC obtained in this study is lower than that obtained by *Camellia sinensis* (L.) O. Kuntze (324.8–528.8 µg/g DW) [92]. These differences can be attributed to the use of different quantification methods, the intrinsic characteristics of varieties, environmental conditions, geographical location, and harvesting period, among other factors [11].

Concerning total flavonoids, contents ranged from 566.17 to 1081.44 µg quercetin/mL extract for all camellias studied (Table 4), showing some significant differences (*p* < 0.05). DD had the highest TFC (1081.44 µg quercetin/mL extract), while GS had the lowest (566.17 µg quercetin/mL extract). Ethanol extracts of fermented *C. japonica* leaves contained a high concentration of TFC (205.19 mg rutin/g) [93]. *C. sinensis* mature leaves showed a high level of TFC (57.39 mg quercetin/g sample), as well [94]. *Camellia chrysantha* also has high TFC content (3975 µg quercetin/g DW) [95].

Regarding TAC, anthocyanins were not detected in GS and CT, which corresponds to flowers of the white ecotype. TAC in flowers for the pink ecotype ranged from 20.57 to 1820.45 μg cyanidin-3-glycoside/g sample, with significant differences (*p* < 0.05) between varieties (Table 4). EM had the highest TAC (1820.45 g cyanidin-3-glycoside/g sample), corresponding to flowers with more intense pink colors (Figure 1). DT and HA obtained the lowest TAC with no statistical differences (<24 g cyanidin-3-glycoside/g sample), which corresponds to pink blush flowers. According to Fan, TAC of *C. japonica* of different ecotypes ranged from 3.83 to 149.34 mg anthocyanin/100 mg [96]. Similar results were reported by Xiaowei, with an average content of anthocyanins of 0.828 mg anthocyanin/g sample [97]. TACs of five *C. japonica* cultivars with different petal colors ranged from 85.09 mM/g to 2.02 mmol/g [98]. 

According to other authors, the main anthocyanin responsible for the red pigmentations of petals was cyanidin 3-*O*-β-glucoside. Additionally, the proportion of cyanidin 3-*O*-β-glucoside and cyanidin-3-*O*-(6-*O*-(E)-*p*-coumaroyl)-B-glucoside was found to be significantly related to the red petal phenotype [96]. As a result, the anthocyanin content allows us to distinguish between the various camellia varieties under consideration. 

The content of phenolic compounds, carotenoids, flavonoids, and anthocyanins in various plant-based foods is closely linked to their antioxidant activity. This is due to these bioactive components exhibiting potent antioxidant properties, effectively combating free radicals and safeguarding cells against oxidative stress [99,100,101]. Numerous studies have shown a positive correlation between higher concentrations of these compounds and increased antioxidant activity [102,103]. Hence, the presence of phenolic compounds, carotenoids, flavonoids, and anthocyanins plays a potential role in promoting overall health benefits and mitigating the risk of chronic diseases.

## 4. Conclusions

Flowers are becoming increasingly popular in the human diet due to consumers’ demand for healthier and more natural products. For this reason, many flowers have been studied in the last decades to explore their nutritional composition and functional properties, particularly their phenolic compound contents and antioxidant properties. However, to the best of our knowledge, there are no reports of *C. japonica* nutritional composition. According to our results, the lipid content of all camellias analyzed were lower than other parts of the plant analyzed in previous studies. All varieties show a relevant content of PUFAs (61–71% of total FAs), with linoleic acid as the most abundant compound in all cases. The protein content of the camellias studied ranged from 4.43 to 6.34 g/100 g DW with not statistically significant differences. *C. japonica* flowers are also a relevant source of minerals like Ca, K, Mg, P, Na, Fe, and Mn, with varieties DT and HA standing out for Ca and P content, respectively (2544.5 and 1566.0 mg/kg). According to the thermochemical characterization and elemental composition of camellias, this raw material has a low ash content, high volatile matter and carbon-fixation rates, and low nitrogen content, making it a promising alternative for biofuel production. Moreover, camellias can be a source of bioactive compounds due to their high content of phenolic compounds (78.45–108.64 mg GAE/g DW), carotenoids (EM > 70 µg/g DW), and total flavonoid content (DD 1081.44 µg quercetin/g DW). The analyzed samples also exhibited significant concentrations of anthocyanins (20.57 to 1820.45 g cyanidin-3-glycoside/g sample), except for GS and CT, where anthocyanins were not detected. These results suggest that *C. japonica* flowers could be a natural source of polyphenols for functional foods, but their true contribution to overall in vivo antioxidant activity is still unknown. 

## Figures and Tables

**Figure 1 foods-12-02825-f001:**
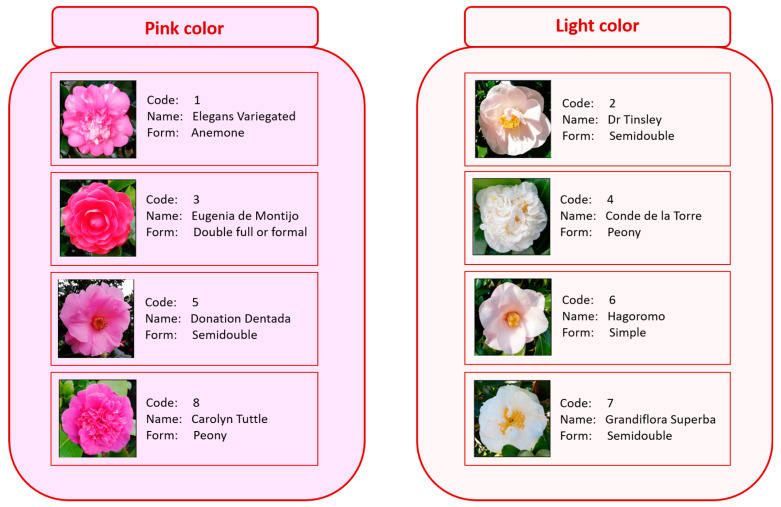
Varieties of *C. japonica* under study.

**Table 1 foods-12-02825-t001:** Nutritional composition and fatty acid profile of 8 varieties of flowers of *C. japonica*.

	Camellia								
	EV	DT	EM	CT	DD	HA	GS	TU	*p*-Value
**Nutritional composition** (g/100 g dry weight)									
Protein	6.3 ± 0.3 ^a^	4.4 ± 0.1 ^c^	4.7 ± 0.1 ^bc^	6 ± 1 ^abc^	5.7 ± 0.1 ^abc^	5.7 ± 0.5 ^abc^	6.1 ± 0.4 ^ab^	5.4 ± 0.4 ^abc^	9.93 × 10^−3^
Lipids	1.2 ± 0.01 ^c^	1.5 ± 0.1 ^ab^	1.2 ± 0.01 ^c^	1.3 ± 0.1 ^bc^	1.6 ± 0.1 ^a^	1.4 ± 0.1 ^abc^	0.9 ± 0.1 ^d^	1.2 ± 0.1 ^c^	9.02 × 10^−7^
**Fatty acid composition** (% total lipids)									
Saturated FAs	29.28 ± 1.1	34.38 ± 0.3	33.48 ± 13.8	25.62 ± 0.6	32.78 ± 0.6	26.08 ± 0.5	33.90 ± 2.3	24.39 ± 0.5	0.33
C14:0 Myristic acid	1.3 ± 0.2 ^ab^	1.2 ± 0.2 ^ab^	1.1 ± 0.3 ^ab^	0.79 ± 0.04 ^b^	1.33 ± 0.04 ^ab^	1.1 ± 0.1 ^ab^	1.7 ± 0.3 ^a^	0.72 ± 0.01 ^b^	0.01
C16:0 Palmitic acid	21.8 ± 0.1	22 ± 1	25 ± 10	19 ± 1	22 ± 1	16 ± 1	20.2 ± 0.4	19 ± 1	0.39
C18:0 Stearic acid	2.0 ± 0.4	2.4 ± 0.2	2 ± 1	1.8 ± 0.3	3.1 ± 0.3	2.2 ± 0.2	2.8 ± 0.4	1.7 ± 0.1	0.07
C20:0 Arachidic acid	1.4 ± 0.1	1.7 ± 0.1	2 ± 1	1.52 ± 0.01	2.0 ± 0.4	2.1 ± 0.3	2.5 ± 0.5	1.83 ± 0.03	0.62
C22:0 Behenic acid	2.8 ± 0.4 ^bc^	7.30 ± 0.02 ^a^	3 ± 1 ^bc^	2.4 ± 0.3 ^bc^	4 ± 1 ^bc^	5 ± 1 ^ab^	7 ± 1 ^a^	1.94± 0.04 ^c^	2.66 × 10^−4^
**Monounsaturated FAs**	4.3 ± 0.3 ^ab^	4.2 ± 0.3 ^ab^	4.1 ± 0.3 ^ab^	3.8 ± 0.2 ^b^	4.7 ± 0.1 ^a^	4.2 ± 0.2 ^ab^	3.9 ± 0.1 ^a^	4.6 ± 0.1 ^ab^	0.03
C20:1 cis-11-Eicosenoic acid	1.6 ± 0.1	1.4 ± 0.1	1.7 ± 0.4	1.4 ± 0.1	1.7 ± 0.1	1.8 ± 0.1	nd	1.89 ^a^ ± 0.03	0.24
C18:1 cis-Oleic acid	2.6 ± 0.4	2.8 ± 0.3	2 ± 1	2.4 ± 0.2	3.0 ± 0.2	2.4 ± 0.3	4 ± 1	2.7 ± 0.2	0.16
**Polyunsaturated FAs**	66.5 ± 1.4	61.0 ± 0.1	62.4 ± 14.2	70.6 ± 0.4	62.5 ± 0.7	69.8 ± 0.7	61.5 ± 2.2	71.1 ± 0.6	0.33
C18:2 cis-Linoleic acid (ω-6)	45 ± 1	44.3 ± 0.3	48 ± 9	51.8 ± 0.2	45.95 ± 0.04	46 ± 1	46 ± 2	52.93 ^a^ ± 0.04	0.19
C18:3 Linolenic acid (ω-3)	19.60 ± 0.03 ^ab^	16.7 ± 0.4 ^ab^	13 ± 5 ^b^	18.0 ± 0.5 ^ab^	17 ± 1 ^ab^	22.7 ± 0.1 ^a^	15.9 ± 0.5 ^ab^	16.82 ± 0.56 ^ab^	0.02
C20:2 cis-11,14-Eicosadienoic acid (ω-6)	1.62 ± 0.02 ^a^	nd	1.15 ± 0.10 ^ab^	0.9 ± 0.1 ^ab^	nd	1.13± 0.04 ^ab^	nd	1.31 ± 0.01 ^a^	4.64 × 10^−4^
**PUFAs ω6**	46.9 ± 1.3	44.8 ± 0.3	49.5 ± 9.1	52.6 ± 0.1	46.0 ± 0.1	47.0 ± 0.9	45.6 ± 1.7	54.2 ± 0.1	0.15
**PUFAs ω3**	19.6 ± 0.1 ^ab^	16.7 ± 0.4 ^ab^	12.9 ± 5.1 ^b^	17.9 ± 0.5 ^ab^	16.6 ± 0.7 ^ab^	22.7 ± 0.2 ^a^	15.9 ± 0.5 ^ab^	16.8 ± 0.6 ^ab^	0.02
**Ratio ω6/ω3**	2.4 ± 0.1 ^b^	2.7 ± 0.1 ^b^	3.8 ± 0.9 ^a^	2.9 ± 0.1 ^ab^	2.8 ± 0.1 ^ab^	2.1 ± 0.1 ^b^	2.9 ± 0.1 ^ab^	3.2 ± 0.1 ^ab^	8.52 × 10^−3^

**FAs**, fatty acids; **EV**, Elegans variegated; **DT**, Dr. Tinsley; **EM**, Eugenia de Montijo; **CT**, Conde de la Torre; **DD**, Donation dentada; **HA**, Hagoromo; **GS**, Grandiflora Superba; **TU**, Carolyn Tuttle; **nd**, not detected. Different letters within the same column indicate statistically significant differences (α = 0.05), according to one-way ANOVA.

**Table 2 foods-12-02825-t002:** Comparative nutritional composition, fatty acid profile, and mineral content of different crops against 8 varieties of flowers of *C. japonica*.

	OB	TV	SO	PO	MI	LE	PA	CJ
**Nutritional composition (%)**								
Proteins	22.53	15.38	9.56	1.57	0.82	0.89	20.8	4.4–6.3
Lipids	1.51	2.09	3.15	0.157	1.36	0.35	58.9	0.9–1.6
**Fatty acid composition (% total lipids)**								
**Saturated FAs**	40.52	43.86	38.79	27.58	79.74	15.10	11.11	24.39–34.38
C14:0 Myristic acid	5.20	21.41	0.43	0.73	nd	nd	nd	0.72–1.7
C16:0 Palmitic acid	22.98	17.12	21.38	9.81	64.42	10.3	9.3	16–25
C18:0 Stearic acid	8.16	3.06	4.10	2.52	9.55	1.6	1.8	1.7–2.8
C20:0 Arachidic acid	nd	nd	nd	nd	0.63	nd	nd	1.4–2.1
C22:0 Behenic acid	nd	nd	nd	nd	nd	nd	nd	1.94–7.30
**Monounsaturated FAs**	21.99	11.98	19.70	5.89	18.62	2.9	70.0	3.8–4.7
C20:1 cis-11-Eicosenoic acid	nd	nd	nd	nd	nd	nd	nd	nd–1.89
C18:1 cis-Oleic acid	17.85	7.54	12.65	5.29	9.01	2.3	69.7	2.0–4.0
**Polyunsaturated FAs**	36.57	43.19	40.96	66.53	1.65	82.0	18.9	61.0–71.1
C18:2 cis-Linoleic acid (ω-6)	17.36	12.62	11.40	11.40	1.65	81.1	18.2	44.3–52.93
C18:3 Linolenic acid (ω-3)	15.95	27.96	12.61	54.92		0.1	0.7	13.0–22.7
C20:2 cis-11,14-Eicosadienoic acid (ω-6)	nd	0.17	3.21	nd	nd	nd	nd	nd–1.62
**Macroelemental composition of the studied camellias (mg/kg).**								
Ca	nd	nd	nd	43.55	11.0	6.26	0.23	1482.0–2544.5
K	nd	nd	nd	33.05	168.0	50.82	nd	6557.0–10,422.0
Mg	nd	nd	nd	nd	10.00	10.46	nd	730.0–1172.0
Na	nd	nd	nd	14.44	1.0	nd	nd	221.0–419.5
P	nd	nd	nd	nd	14.0	228.5	0.49	1091.0–1566.0
**Microelemental composition of the studied camellias (mg/kg).**								
Fe	624.51	690.05	732.72	4.44	0.16	nd	3.5	13.59–33.49
Mn	78.46	96.11	68.92	nd	nd	nd	nd	34.67–214.21
Zn	54.63	31.73	38.87	2.08	0.09	nd	nd	4.57–9.58
Cu	27.69	7.41	7.89	nd	0.04	nd	nd	3.54–6.24
Se	nd	nd	nd	nd	0.6	nd	nd	0.11–0.17
**Reference**	[44]	[44]	[44]	[45,46]	[47,48]	[49,50]	[51,52]	This study

**FAs**, fatty acids; **OB**, *Ocimum basilicum* (basil); **TV**, *Thymus vulgaris* (thyme); **SO**, *Salvia officinalis* (sage); **PO**, *Portulaca oleracea* L. (purslane); **MI**, *Mangifera indica* L. (mango); **LE**, *Lentinula edodes* (shiitake); **PA**, *Prunus amygdalus* “dulcis” (almond); **CJ**, *C. japonica* (range of the different varieties of this study); **nd**, not detected.

**Table 3 foods-12-02825-t003:** Macro-, micro-, and basic elemental composition of the studied camellias.

	Camellia								
	EV	DT	EM	CT	DD	HA	GS	TU	*p*-Value
**Macroelemental composition of the studied camellias (mg/kg).**									
**Ca**	1693.0 ± 145.7 ^de^	2295.5 ± 47.4 ^b^	1482.0 ± 21.2 ^e^	2067.5 ± 5.0 ^bc^	1852.5 ± 17.7 ^c^	2544.5 ± 44.6 ^a^	1633.5 ± 31.8 ^de^	1822.5 ± 0.7 ^d^	1.17 × 10^−6^
**K**	9968.5 ± 27.6 ^ab^	9056.5 ± 61.5 ^bc^	7570.5 ± 279.3 ^de^	10,322.5 ± 726.2 ^a^	7985.0 ± 297.0 ^cd^	9646.5 ± 64.4 ^ab^	10,422.0 ± 116.0 ^a^	6557.0 ± 76.4 ^e^	8.56 × 10^−6^
**Mg**	994.5 ± 7.8 ^bc^	1156.0 ± 14.1 ^a^	730.0 ± 33.9 ^d^	1017.5 ± 46.0 ^b^	904.0 ± 5.7 ^c^	1020.5 ± 40.3 ^b^	1172.0 ± 9.9 ^a^	934.0 ± 33.9 ^bc^	5.49 × 10^−6^
**Na**	259.5± 5.0 ^cd^	419.5 ± 23.2 ^a^	415.0 ± 35.4 ^ab^	337.0 ± 4.2 ^abc^	221.0 ± 42.4 ^d^	312.5 ± 26.2 ^bcd^	286.5 ± 6.4 ^cd^	250.5 ± 31.8 ^cd^	3.80 × 10^−4^
**P**	1566.0 ± 36.8 ^a^	1219.5 ± 27.6 ^cd^	1145.0 ± 28.3 ^d^	1403.0 ± 33.4 ^b^	1463.5 ± 58.7 ^ab^	1536.5 ± 20.5 ^a^	1347.0 ± 26.9 ^bc^	1091.0 ± 1.4 ^d^	3.13 × 10^−6^
**Microelemental composition of the studied camellias (mg/kg).**									
**Fe**	19.87 ± 2.32 ^ab^	21.76 ± 0.48 ^ab^	13.59 ± 0.57 ^b^	18.57 ± 1.49 ^ab^	19.10 ± 0.93 ^ab^	26.26 ± 12.32 ^ab^	33.49 ± 3.25 ^a^	15.89 ± 0.57 ^ab^	0.04
**Mn**	40.92 ± 1.72 ^e^	175.73 ± 7.65 ^b^	43.65 ± 0.08 ^e^	102.34 ± 1.90 ^d^	152.655 ± 8.75 ^c^	34.67 ± 1.74 ^e^	146.20 ± 0.33 ^c^	214.21 ± 6.36 ^a^	1.41 × 10^−9^
**Zn**	4.57 ± 0.30 ^e^	9.58 ± 0.16 ^a^	5.15 ± 0.16 ^de^	5.85 ± 0.45 ^cd^	8.33 ± 0.49 ^b^	8.52 ± 0.01 ^ab^	6.62 ± 0.08 ^c^	5.67 ± 0.08 ^cd^	6.37 × 10^−7^
**Cu**	5.91 ± 0.11 ^a^	3.92 ± 0.03 ^b^	3.72 ± 0.17 ^b^	5.39 ± 0.39 ^a^	4.14 ± 0.18 ^b^	6.24 ± 0.35 ^a^	5.53 ± 0.18 ^a^	3.54 ± 0.11 ^b^	7.09 × 10^−6^
**Se**	0.13 ± 0.06	0.12 ±0.01	0.12 ± 0.04	0.17 ± 0.04	0.12 ± 0.05	0.15 ± 0.04	0.11 ± 0.05	0.14 ± 0.03	0.91
**Thermal analysis (%)**									
**Moisture**	3.82 ± 0.51 ^b^	5.55 ± 0.33 ^b^	5.96 ± 2.22 ^b^	4.78 ± 0.40 ^b^	4.59 ± 0.09 ^b^	4.12 ± 0.37 ^b^	9.91 ± 0.23 ^a^	5.45 ± 0.44 ^b^	2.08 × 10^−3^
**Ash**	1.75 ± 1.63	2.60 ± 0.01	3.80 ± 1.56	2.90 ± 0.42	2.35 ± 0.92	2.50 ± 0.28	3.30 ± 0.71	2.00 ± 0.42	0.47
**Volatile**	73.53 ± 2.50 ^a^	70.97 ± 0.44 ^a^	69.88 ± 0.50 ^a^	71.11 ± 0.20 ^a^	70.77 ± 0.51 ^a^	72.83 ± 0.11 ^a^	65.03 ± 0.38 ^b^	72.79 ± 0.03 ^a^	3.73 × 10^−4^
**Fixed C**	20.91 ± 0.32 ^bc^	20.65 ± 0.39 ^bc^	20.54 ± 0.03 ^bc^	20.27 ± 0.27 ^c^	22.33 ± 0.28 ^a^	20.49 ± 0.67 ^bc^	21.75 ± 0.09 ^ab^	19.74 ± 0.07 ^c^	9.39 × 10^−4^
**Elemental composition (%)**									
**N**	1.02 ± 0.05 ^a^	0.71 ± 0.02 ^c^	0.75 ± 0.02 ^bc^	0.93 ± 0.10 ^abc^	0.92 ± 0.02 ^abc^	0.92 ± 0.09 ^abc^	0.98 ± 0.06 ^ab^	0.86 ± 0.07 ^abc^	0.01
**C**	44.09 ± 0.29 ^bc^	44.22 ± 0.20 ^b^	43.18 ± 0.14 ^cd^	44.10 ± 0.09 ^bc^	45.78 ± 0.43^a^	44.81 ± 0.02 ^b^	42.27 ± 0.28 ^d^	43.98 ± 0.23 ^bc^	1.83 × 10^−5^
**H**	6.39 ± 0.04	6.45 ± 0.17	6.34 ± 0.19	6.47 ± 0.28	6.35 ± 0.64	6.49 ± 0.08	6.25 ± 0.04	6.55 ± 0.10	0.95
**O**	48.50 ± 0.30 ^bc^	48.62 ± 0.38 ^bc^	49.74 ± 0.35 ^ab^	48.51 ± 0.08 ^bc^	46.97 ± 1.06^c^	47.78 ± 0.01^c^	50.52 ± 0.18 ^a^	48.61 ± 0.40 ^bc^	1.38 × 10^−3^
**HHV**(MJ/kg)	21.32	21.49	21.26	21.17	21.61	21.18	22.33	21.36	

**EV**, Elegans variegated; **DT**, Dr. Tinsley; **EM**, Eugenia de Montijo; **CT**, Conde de la Torre; **DD**, Donation dentada; **HA**, Hagoromo; **GS**, Grandiflora Superba; **TU**, Carolyn Tuttle. Different letters within the same column indicate statistically significant differences (α = 0.05), according to one-way ANOVA.

**Table 4 foods-12-02825-t004:** Bioactive compound composition and antioxidant activity of the studied camellias.

Camellia	YIELD	TPC	TCC	TFC	TAC
	(%)	(mg GAE/g DW)	(µg C/g DW)	(µg Q/g DW)	(µg cyd/g DW)
**EV**	61.23 ± 0.80 ^ab^	95.73 ± 20.41 ^ab^	33.35 ± 1.89 ^d^	782.12 ± 174.69 ^bc^	991.96 ± 25.94 ^c^
**DT**	58.96 ± 2.01 ^bc^	104.79 ± 22.41 ^a^	26.03 ± 3.49 ^ef^	703.98 ± 132.7 ^bc^	20.57 ± 6.60 ^e^
**EM**	61.87 ± 0.96 ^ab^	96.57 ± 22.22 ^ab^	73.21 ± 4.75 ^a^	656.43 ± 151.15 ^bc^	1820.45 ± 147.96 ^a^
**CT**	57.00 ± 2.40 ^cd^	108.64 ± 31.39 ^a^	49.44 ± 1.13 ^c^	796.45 ± 296.97 ^abc^	nd
**DD**	55.14 ± 0.41 ^d^	95.63 ± 30.12 ^ab^	30.71 ± 0.62 ^de^	1081.44 ± 153.03 ^a^	400.61 ± 10.06 ^d^
**HA**	62.02 ± 2.28 ^ab^	103.13 ± 19.38 ^a^	24.80 ± 1.59 ^f^	855.00 ± 239.48 ^abc^	23.00 ± 4.98 ^e^
**GS**	62.68 ± 2.05 ^a^	93.35 ± 11.16 ^ab^	28.85 ± 2.00 ^def^	566.17 ± 78.82 ^c^	nd
**TU**	61.61 ± 0.79 ^ab^	78.45 ± 13.00 ^b^	63.90 ± 1.87 ^b^	892.96 ± 257.13 ^ab^	1230.53 ± 45.44 ^b^
** *p* ** **-value**	4.01 × 10^−8^	3.44 × 10^−3^	2.00 × 10^−16^	8.89 × 10^−8^	1.06 × 10^−18^

**TPC**, total phenolic content; **TCC**, total carotenoid content; **TFC**, total flavonoid content; **TAC**, total anthocyanin content; **GAE**, gallic acid equivalents; **Q**, quercetin; **E**, extracts; cyd, cyanidin-3-glycoside; **EV**, Elegans variegated; **DT**, Dr. Tinsley; **EM**, Eugenia de Montijo; **CT**, Conde de la Torre; **DD**, Donation dentada; **HA**, Hagoromo; **GS**, Grandiflora Superba; **TU**, Carolyn Tuttle; **nd**, not detected. Different letters within the same column indicate statistically significant differences (α = 0.05), according to one-way ANOVA.

## Data Availability

All related data are presented in this paper. Additional inquiries should be addressed to the corresponding author.

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
