# Peer review of "Camellia japonica Flowers as a Source of Nutritional and Bioactive Compounds"

_foods, 2023, doi:10.3390/foods12152825_

Round 1
Reviewer 1 Report
The current paper studied the nutritional and bioactive compounds determined in Camellia japonica flowers.
The following points should be addressed:
1. For statistical analyses, how many repetitions were considered for each type of flower? The number of observations needs to be added to the material and methods, as well under each table.
2. All tables should respect the journal format. All the values in the tables should be presented as mean and the SEM and p values to be on separate columns. However, if there are under 6 no of observation, the p-value should be excluded, because it will not be relevant.
3. Please use decimals in all values presented in the tables for better data uniformity.
4. The values presented in Table 2 are very small and is hard to follow them, please revise them during the revision stage.
5. Since Camellia are plants, their chemical composition, mineral, and antioxidant compounds should be discussed a little also back to back with other plants that have gained the attention of researchers (for example, basil, sage, garlic etc). How Camellia species can be superior to these other well-studied plants rich in bioactive compounds?
6. Please proofread the paper and correct small typos and grammatical errors that can change the meaning of some sentences.
Overall, the paper is nicely written.
Author Response
The current paper studied the nutritional and bioactive compounds determined in Camellia japonica flowers.
The following points should be addressed:
- For statistical analyses, how many repetitions were considered for each type of flower? The number of observations needs to be added to the material and methods, as well under each table.
Answer: All flower varieties in this study were subjected to an equal number of repetitions in each trial conducted. In the “Evaluation of Camellias as a Promising Source of Nutrients” section (section 2.2), it should be noted that all tests were conducted in duplicate. This additional information has been included to provide a comprehensive understanding of the experimental design and ensure transparency in reporting. However, it is important to note that the number of determinations varied for each specific test within section 2.3. (“Determination of bioactive compounds from C. japonica L. flowers”). The details regarding the number of determinations for each test are specified within the methods of the bioactive compounds section, accounting for the differences observed (i.e., determinations were done fivehold, determinations were conducted quadrice, etc.).
- All tables should respect the journal format. All the values in the tables should be presented as mean and the SEM and p values to be on separate columns. However, if there are under 6 no of observation, the p-value should be excluded, because it will not be relevant.
Answer: We understand your concern regarding the inclusion of p-values when the sample size is less than six observations. However, we would like to clarify our rationale for including these values in our study. While we acknowledge that the conventional practice is to report p-values only when there are more than six observations, we made a deliberate decision to provide complete information in our research. By including the p-values in separate columns, we aim to ensure transparency and facilitate further analysis or meta-analysis by interested researchers. We understand that this approach might deviate from the standard practice, and we are open to considering alternative ways of presenting our findings if it is deemed more appropriate by the scientific community. We appreciate your understanding and valuable feedback, and we will carefully consider your suggestions for improving the manuscript. Thank you once again for your time and effort in reviewing our work.
- Please use decimals in all values presented in the tables for better data uniformity.
Answer: We appreciate your suggestion to use decimals for better data uniformity in all values presented in the tables. However, after careful consideration, we have decided not to implement this change. Allow us to explain our reasoning behind this decision. The number of decimal places is determined by the magnitude of the standard deviation associated with each measurement. Therefore, if the error is in the units rango, there is no meaning in presenting decimal places. We believe that presenting decimal places without proper significance may give a false impression of precision, especially when the error lies within the range of the units being measured. In such cases, presenting decimal places would not contribute to a meaningful interpretation of the data.
- The values presented in Table 2 are very small and is hard to follow them, please revise them during the revision stage.
Answer: We have taken your suggestion into consideration and have now enlarged the size of the numbers to ensure readability. We have carefully assessed the readability and believe that the current size is adequate for clear comprehension without causing any difficulty.
- Since Camellia are plants, their chemical composition, mineral, and antioxidant compounds should be discussed a little also back to back with other plants that have gained the attention of researchers (for example, basil, sage, garlic etc). How Camellia species can be superior to these other well-studied plants rich in bioactive compounds?
Answer: We would like to inform you that we have addressed your concern by adding a table in the manuscript that compares the obtained ranges from the different varieties of C. japonica of our study with previous studies on other plants such as basil, sage, and garlic. This table has been thoroughly discussed in the article, highlighting the similarities and differences observed in the nutritional composition, fatty acid profile and mineral content among these plants.
By including this comparative table and discussing it in the article, we aim to provide a comprehensive analysis of the nutritional characteristics of C. japonica in relation to other well-studied plants. This comparison allows for a better understanding of the unique nutritional profile of C. japonica and its potential implications.
- Please proofread the paper and correct small typos and grammatical errors that can change the meaning of some sentences.
Answer: Thank you for your message. We would like to inform you that the English language in our manuscript has been thoroughly reviewed and revised by a native English speaker. Their expertise ensured that the paper adheres to the highest linguistic standards and maintains clarity throughout.
We greatly appreciate your vigilance in maintaining the quality of the language used in scientific manuscripts. Should you have any further suggestions or specific areas of concern regarding the language, please feel free to share them with us. We are dedicated to addressing any remaining issues and ensuring the manuscript is of the utmost quality.
Once again, we would like to express our gratitude for your valuable feedback and guidance. Your contributions are instrumental in improving the overall quality of our work.
Overall, the paper is nicely written.
Answer: Thank you for your kind comment on the writing quality of our paper. We appreciate your positive feedback and acknowledgment of our efforts to ensure clarity and coherence in presenting our research findings.
Reviewer 2 Report
Manuscript titled “Camellia japonica flowers as a source of nutritional and bioactive compounds“reports the nutritional and bioactive compounds composition of eight varieties of C. japonica L. flowers. There are some comments for the authors.
1. Line 20: Please add major ingredient content data to the abstract:
2.The introduction needs to be improved. Re-write it with proper hypotheses before objectives and review the literature properly.
3. Line 83: Please add the freeze-drying conditions.
4. Line 92: Leave a space before °C. Please check whole paper.
5. Line 94: use SI units only in your paper. Note that the form g 100g ^-1, etc. is not correct. Avoid the use of g per 100g, for example in food/feed composition, by using g kg ^-1.
6. Line 192: use xg values and not rpm.
7. Discussion is needed to be improved with a good review of the literature. Rewrite it with mechanistic explanations and interpretations of results chronologically citing proper pieces of literature.
8. Line 381: Functional evaluation should not only include ingredients, please add specific evaluation indicators according to the characteristics of ingredients, such as antioxidant, etc
9. Line 448-470: Conclusion: This section needs to be improved.
10. Please consider having your manuscript revised to amend various typos and writing mistakes. This should ideally be done by a native English-speaking colleague or professional service.
Please consider having your manuscript revised to amend various typos and writing mistakes. This should ideally be done by a native English-speaking colleague or professional service.
Author Response
Manuscript titled “Camellia japonica flowers as a source of nutritional and bioactive compounds“reports the nutritional and bioactive compounds composition of eight varieties of C. japonica L. flowers. There are some comments for the authors.
- Line 20: Please add major ingredient content data to the abstract:
Answer: Main numerical data results were added to the abstract.
“Our results show that camellias have a high nutritional value, with carbohydrates as the most abundant macronutrients, followed by a moderate protein content (4.4—6.3 g / 100g dry weight) and high levels of polyunsaturated fatty acids (especially ω-3 fatty acids, which represent 12.9 – 22.7 % of the total fatty acids), raising its potential use for nutritional purposes.”
2.The introduction needs to be improved. Re-write it with proper hypotheses before objectives and review the literature properly.
Answer: Based on your suggestions and the gaps identified in our initial submission, we conducted a comprehensive literature search and incorporated additional relevant references. These new sources have bolstered the supporting evidence and further strengthened the arguments presented in our research.
- Line 83: Please add the freeze-drying conditions.
Answer: We have carefully considered your comments and have made the necessary revisions to address the concerns raised. As you can observe in the following paragraph, we have added the requested information, ensuring that all the relevant aspects are now adequately covered.
“After collection, flowers were frozen at -80 °C and lyophilized (LyoAlfa10/15, Telstar, Thermo Fisher Scientific) under specific conditions: the freezing time in the equipment was set to 1 hour, fol-lowed by a warming process for 20 minutes using a back-vacuum (BV) method. The main drying phase lasted for 2 hours and 30 minutes at a pressure of 0.01000 mbar, while the final drying stage extended for 72 hours at a pressure of 0.0010 mbar. Once lyophilized, the flowers were crushed into a fine powder (1.8 mm mesh), and stored at -20 °C until analysis.”
- Line 92: Leave a space before °C. Please check whole paper.
Answer: Thank you for bringing the mistake to our attention. We sincerely apologize for the oversight. We have made the necessary corrections by adding a space before each temperature unit throughout the document.
- Line 94: use SI units only in your paper. Note that the form g 100g ^-1, etc. is not correct. Avoid the use of g per 100g, for example in food/feed composition, by using g kg ^-1.
Answer: Thank you for your feedback. We acknowledge your concern regarding the units used in our manuscript. While we understand that the International System of Units (SI) is the standard for scientific publications, we would like to explain our rationale for using the units that are most commonly employed in scientific studies, as reflected in the new table.
By utilizing the frequently used units, we aim to enhance the comparability of our findings with other related studies in the field. This consistency in units allows for easier cross-referencing and comparison of data across different research works. Furthermore, it ensures that our results can be readily understood and applied by researchers who are accustomed to these units.
- Line 192: use xg values and not rpm.
Answer: Thank you for notifying us about the change in measurement units. We apologize for any confusion caused. We have duly updated the document to reflect the revised measurement units.
- Discussion is needed to be improved with a good review of the literature. Rewrite it with mechanistic explanations and interpretations of results chronologically citing proper pieces of literature.
Answer: In response to your comment, we have thoroughly revised the Discussion section, incorporating mechanistic explanations and interpretations of our findings. We have taken into account the chronological order of the studies cited, ensuring that proper references are provided to support our claims.
To ensure the completeness of our literature review, we extensively utilized high-quality bibliographic resources on Camellias that were available to us. Additionally, we have incorporated studies from other related crops to provide meaningful comparisons and contextualize our findings within a broader scientific framework.
By incorporating mechanistic explanations and expanding our literature review, we aim to provide a more robust discussion that elucidates the significance of our results and their implications within the existing scientific knowledge.
- Line 381: Functional evaluation should not only include ingredients, please add specific evaluation indicators according to the characteristics of ingredients, such as antioxidant, etc.
Answer: We appreciate your suggestion to expand the functional evaluation beyond just the ingredients. We completely agree with your point, and we understand the importance of incorporating specific evaluation indicators based on the characteristics of the ingredients. To address this concern, we revised the manuscript by including specific evaluation indicators, such as antioxidant activity, in accordance with the respective ingredient characteristics. By incorporating these indicators, we aim to provide a more comprehensive and detailed functional evaluation. We appreciate your guidance in improving the quality and relevance of our research. Your valuable input has allowed us to further refine our manuscript and enhance its overall contribution to the scientific community.
- Line 448-470: Conclusion: This section needs to be improved.
Answer: Thank you for your feedback on the conclusion section of our scientific article. We appreciate your comment and understand the importance of providing a strong and well-crafted conclusion. To address this concern, we thoroughly revisited the conclusion section and made necessary improvements. We have ensured that the conclusion effectively summarizes the key findings and highlights the significance of our research. Additionally, we tried to provide a clear and concise summary of the implications and potential future directions based on our results. Thank you once again for your valuable feedback. We are committed to delivering a comprehensive and impactful conclusion that aligns with the overall quality of our research.
- Please consider having your manuscript revised to amend various typos and writing mistakes. This should ideally be done by a native English-speaking colleague or professional service.
Answer: Thank you for your message. We are pleased to inform you that the manuscript has undergone a comprehensive review by a native English speaker. Their expertise ensured the accuracy and clarity of the language used throughout the paper.
We genuinely appreciate your commitment to maintaining the high standards of language in scientific manuscripts. If you have any specific suggestions or concerns regarding the language, we would be more than happy to address them. Our goal is to ensure the manuscript meets the highest quality standards in terms of language and clarity.
Once again, we sincerely thank you for your valuable feedback and guidance. Your contributions are invaluable in enhancing the overall quality of our work.
Reviewer 3 Report
Article: Camellia japonica flowers as a source of nutritional and bioactive compounds
The authors describe in detail the methodology of performing laboratory analyses, but the description of statistical methods comes down to a short list. ANOVA model, number of repetitions is not given. The presentation of numerical values in Tab. 1 and Tab. 2 is incorrect. Values of the same variable are presented with different accuracy (example: C16:0 Palmitic acid 21.8±0.1 22±1 25±10). The assignment of the letters of detailed comparisons is probably accidental and not based on detailed comparisons. Example Table 3. The TPC variable, group letter "a" is assigned for the minimum and maximum value, and there is also an inclusive group "ab". The conclusions can be extended, the research material is very rich and not fully discussed.
Author Response
The authors describe in detail the methodology of performing laboratory analyses, but the description of statistical methods comes down to a short list. ANOVA model, number of repetitions is not given.
Answer: All flower varieties in this study were subjected to an equal number of repetitions in each trial conducted. In the “Evaluation of Camellias as a Promising Source of Nutrients” section (section 2.2), it should be noted that all tests were conducted in duplicate. This additional information has been included to provide a comprehensive understanding of the experimental design and ensure transparency in reporting. However, it is important to note that the number of determinations varied for each specific test within section 2.3. (“Determination of bioactive compounds from C. japonica L. flowers”). The details regarding the number of determinations for each test are specified within the methods of the bioactive compounds section, accounting for the differences observed (i.e., determinations were done fivehold, determinations were conducted quadrice, etc.).
The presentation of numerical values in Tab. 1 and Tab. 2 is incorrect. Values of the same variable are presented with different accuracy (example: C16:0 Palmitic acid 21.8±0.1 22±1 25±10).
Answer: We appreciate your suggestion to use decimals for better data uniformity in all values presented in the tables. However, after careful consideration, we have decided not to implement this change. Allow us to explain our reasoning behind this decision. The number of decimal places is determined by the magnitude of the standard deviation associated with each measurement. Therefore, if the error is in the units rango, there is no meaning in presenting decimal places. We believe that presenting decimal places without proper significance may give a false impression of precision, especially when the error lies within the range of the units being measured. In such cases, presenting decimal places would not contribute to a meaningful interpretation of the data.
The assignment of the letters of detailed comparisons is probably accidental and not based on detailed comparisons. Example Table 3. The TPC variable, group letter "a" is assigned for the minimum and maximum value, and there is also an inclusive group "ab".
Answer: Thank you for your feedback on the assignment of letters for detailed comparisons in the manuscript. We appreciate your observation regarding Table 3, specifically the assignment of group letters "a" and "ab" for the TPC variable.
Upon reviewing the data and carefully examining the issue you raised, we have identified the errors that occurred during the data transfer process. We apologize for any inconvenience caused by these mistakes. We have taken the necessary steps to correct the errors and ensure the accurate representation of the data. The revised version of Table 3 now reflects the appropriate assignments of group letters based on detailed comparisons.
Thank you for bringing this to our attention, and we appreciate your thorough review of our manuscript. If you have any further concerns or suggestions, please don't hesitate to let us know.
The conclusions can be extended, the research material is very rich and not fully discussed.
Answer: Thank you for your feedback on the conclusion section of our scientific article. We appreciate your comment and understand the importance of providing a strong and well-crafted conclusion. To address this concern, we thoroughly revisited the conclusion section and made necessary improvements. We have ensured that the conclusion effectively summarizes the key findings and highlights the significance of our research. Additionally, we tried to provide a clear and concise summary of the implications and potential future directions based on our results. Thank you once again for your valuable feedback. We are committed to delivering a comprehensive and impactful conclusion that aligns with the overall quality of our research.
Reviewer 4 Report
“Camellia japonica flowers as a source of nutritional and bioactive compounds”.
This is an interesting topic, many nutrients and active substances are determined to prove that camellias can be utilized and developed into many products, including functional ingredients, biofuel production, and bioactive compounds.
There was much laboratory work. But the thing that is most concerning to me (and perhaps other scientists) is the lack of sampling details in this manuscript. Because plants are very sensitive to environmental conditions, "harvested in Galicia during the winter of 2020" is not scientific rigor. Details about the sample collection, like standards for harvest (maturity? leaf color? water content?), detailed sampling sites in Galicia, especially for different varieties, and whether they grew in a similar environment (temperature, precipitation, soil fertility, etc.), and the most important, duplications for sampling, should be clarified.
By highlighting the nutritional value of Camellia, it may be better to compare their factional contents with normal foods like crops or fruits, and in order to improve readability, it may be better to summarize the comparison in a table or figure.
Why were eight varieties of Camellia selected in this study, and how about the production, planting area, or distribution of different varieties?
Author Response
“Camellia japonica flowers as a source of nutritional and bioactive compounds”.
This is an interesting topic, many nutrients and active substances are determined to prove that camellias can be utilized and developed into many products, including functional ingredients, biofuel production, and bioactive compounds.
Answer: Thank you for your insightful comment on our manuscript. We appreciate your recognition of the potential applications and versatility of camellias as a source of nutrients and active substances.
There was much laboratory work. But the thing that is most concerning to me (and perhaps other scientists) is the lack of sampling details in this manuscript. Because plants are very sensitive to environmental conditions, "harvested in Galicia during the winter of 2020" is not scientific rigor. Details about the sample collection, like standards for harvest (maturity? leaf color? water content?), detailed sampling sites in Galicia, especially for different varieties, and whether they grew in a similar environment (temperature, precipitation, soil fertility, etc.), and the most important, duplications for sampling, should be clarified.
Answer: We fully acknowledge the significance of all these factors in the composition of the analyzed flowers. Therefore, we have revised section 2.1 (sample collection) to include the requested information. The updated version is as follows:
“The flowers used in this study were obtained from eight different cultivars of Camellia japonica L. These cultivars were botanically identified by official germplasm banks and identification guide resources. The specific cultivars include in this study were ‘Conde de la Torre’ (CT), ‘Elegans variegated’ (EV), ‘Donation dentada’ (Camellia japonica ´ Camellia saluenensis, DD), ‘Dr. Tinsley’ (DT), ‘Eugenia de Montijo’ (EM), ‘Grandiflora superba’ (GS), ‘Hagoromo’ (HA), and ‘Carolyn Tuttle’ (CT). These cultivars were selected for their distinct characteristics and were used in subsequent analyses and experiments. The collection took place in NW Spain (42.431° N, 8.6444° W) in January 2020, and the collection was carried out by Viveiros Moreira. After collection, flowers were frozen at -80 °C and lyophilized (LyoAlfa10/15, Telstar, Thermo Fisher Scientific) under specific conditions: the freezing time in the equipment was set to 1 hour, fol-lowed by a warming process for 20 minutes using a back-vacuum (BV) method. The main drying phase lasted for 2 hours and 30 minutes at a pressure of 0.01000 mbar, while the final drying stage extended for 72 hours at a pressure of 0.0010 mbar. Once lyophilized, the flowers were crushed into a fine powder (1.8 mm mesh), and stored at -20 °C until analysis.”
Moreover, we have clarified the information regarding the duplication of tests in the respective sections as follow. All flower varieties in this study were subjected to an equal number of repetitions in each trial conducted. In the “Evaluation of Camellias as a Promising Source of Nutrients” section (section 2.2), it should be noted that all tests were carried out in duplicate. This additional information has been included to provide a comprehensive understanding of the experimental design and ensure transparency in reporting. However, it is important to note that the number of determinations varied for each specific test within section 2.3. (“Determination of bioactive compounds from C. japonica L. flowers”). The details regarding the number of determinations for each test are specified within the methods of the bioactive compounds section, accounting for the differences observed (i.e., determinations were done fivehold, determinations were conducted quadrice, etc.).
By highlighting the nutritional value of Camellia, it may be better to compare their factional contents with normal foods like crops or fruits, and in order to improve readability, it may be better to summarize the comparison in a table or figure.
Answer: Thank you for your suggestion. We have taken it into consideration and have made the necessary revisions to the manuscript. We have included a table that compares the nutritional value of Camellia with various crops and fruits, as you recommended. This will help highlight the differences and provide a clearer understanding of the nutritional composition.
We appreciate your keen attention to readability and ensuring that the information is presented in a more concise and accessible manner. The table will indeed enhance the clarity and make the comparison more easily comprehensible for readers.
Why were eight varieties of Camellia selected in this study, and how about the production, planting area, or distribution of different varieties?
Answer: The rationale behind selecting these specific varieties was primarily based on their prevalence and cultivation in the same geographic region. We aimed to minimize the confounding effects of external factors, such as edaphoclimatic conditions, by focusing on varieties that were already adapted to the local environment.
Regarding the production, planting area, and distribution of different varieties, our study primarily focused on assessing the intrinsic variations among the selected Camellia cultivars rather than investigating their broader commercial aspects. While we acknowledge that these factors play an important role in the industry, our research was designed to specifically address the genetic and phenotypic differences among the chosen varieties.
We appreciate your feedback and suggestion to consider additional aspects in future studies, such as the production, planting area, and distribution of Camellia varieties, to provide a more comprehensive understanding of the subject.
Thank you for your valuable questions, which have prompted us to reflect on the scope and potential future directions of our study.